# Physics-Informed Discrepancy Decomposition and Robust Astrophysical Inference for GW231123

## Abstract

Robust astrophysical interpretations from gravitational-wave parameter inference require addressing model-dependent biases. We present a physics-informed framework to decompose discrepancies among five waveform models (NRSur7dq4, IMRPhenomXO4a, SEOBNRv5PHM, IMRPhenomXPHM, IMRPhenomTPHM) for GW231123. Our approach combines exploratory metrics (Jensen-Shannon Divergence, Wasserstein distance), high-dimensional analysis with UMAP, and a Physics-Informed Discrepancy Decomposition. This decomposition quantifies divergences in parameter subspaces—mass/distance, effective spin, individual spin/orientation, and remnant properties—linking model differences to physical approximations. We find substantial disagreements in inferred component masses, effective spin, and redshift, with UMAP separating models into distinct clusters. Discrepancy attribution shows individual spin/orientation is most model-dependent due to spin-precession treatments, while remnant properties reflect merger-ringdown modeling. Crucially, no astrophysical parameter for GW231123 is robust across all models, as systematic waveform uncertainties exceed statistical errors. Thus, for high-mass, precessing binary black hole mergers, waveform choice dominates inference, limiting firm astrophysical conclusions unless model biases are explicitly accounted for.[1]

## 1 Introduction

The advent of gravitational-wave (GW) astronomy, beginning with LIGO-Virgo-KAGRA (LVK) detections of binary black hole (BBH) mergers, has transformed our understanding of energetic astrophysical events. GW signals allow us to infer black hole properties, formation channels, and test spacetime under extreme conditions. Yet, such interpretations critically depend on theoretical waveform models, which approximate Einstein's field equations with varying fidelity—from accurate but costly numerical relativity (NR) simulations to faster semi-analytical and phenomenological models.

High-mass, spinning, and precessing BBH systems particularly require approximate models, since NR is too computationally expensive for large-scale inference. Consequently, multiple waveform families exist, differing in efficiency, higher-order mode inclusion, and spin-precession treatment. This diversity introduces model-dependent biases that can dominate statistical errors, especially for challenging events like GW231123. To achieve robust science, it is essential not only to observe

---

[1]This paper, including the idea and the research analysis, was fully generated and written by Denario, a multi-AI agent system. All the input and output files, together with the original paper, can be found in the supplementary material. The Denario code is available in the supplementary material and a YouTube video demonstrating the end-to-end research pipeline with Denario is available in the anonymised YouTube channel at this link.

Submitted to 1st Open Conference on AI Agents for Science (agents4science 2025). Do not distribute.

model discrepancies but to identify why they arise and which physical aspects of the source are most sensitive.

In this paper, we introduce a physics-informed framework to decompose discrepancies among five models—NRSur7dq4, IMRPhenomXO4a, SEOBNRv5PHM, IMRPhenomXPHM, and IMR-PhenomTPHM [17]. Our key innovation, Physics-Informed Discrepancy Decomposition, defines parameter subspaces (mass/distance, effective spin, individual spin/orientation, remnant properties) and quantifies divergences within them, directly linking differences in inferred parameters to known model approximations such as spin-precession treatment or higher-order mode inclusion.

To validate this approach, we first perform exploratory analysis: computing marginal posterior divergences via Jensen-Shannon Divergence (JSD) and 1-Wasserstein distance, and applying Uniform Manifold Approximation and Projection (UMAP) to reveal model clustering in the high-dimensional parameter space. The physics-informed decomposition then quantifies disagreement within each subspace, identifying which GW231123 properties are robust across models and which remain model-sensitive. This provides interpretable insights into astrophysical inference robustness and enables more reliable consensus constraints for GW231123.

## 2 Methods

### 2.1 Data Acquisition and Pre-processing

Our analysis initiates with the acquisition and meticulous pre-processing of posterior samples derived from the gravitational-wave event GW231123. These samples, representing the probability distributions of various source parameters, were generated using five distinct gravitational-wave waveform models: NRSur7dq4, IMRPhenomXO4a, SEOBNRv5PHM, IMRPhenomXPHM, and IMRPhenomT-PHM. Each model's posterior samples were provided as individual CSV files, specifically located at '/mnt/ceph/users/anonymous_human/AstroPilot/GW/Iteration1/data/GW231123_NRSur7dq4.csv', '/mnt/ceph/users/anonymous_human/AstroPilot/GW/Iteration1/data/GW231123_IMRPhenomXO4a.csv', '/mnt/ceph/users/anonymous_human/AstroPilot/GW/Iteration1/data/GW231123_SEOBNRv5PHM.csv', '/mnt/ceph/users/anonymous_human/AstroPilot/GW/Iteration1/data/GW231123_IMRPhenomXPHM.csv', and '/mnt/ceph/users/anonymous_human/AstroPilot/GW/Iteration1/data/GW231123_IMRPhenomTPHM.csv'.

Upon loading, each CSV file was parsed into a separate pandas DataFrame [26]. To facilitate unified analysis while preserving model attribution, a 'model' column was appended to each DataFrame, explicitly identifying the waveform model from which the samples originated. These individual DataFrames were then consolidated into a single, master Python dictionary, with model names serving as keys, providing a structured and accessible representation of the entire dataset.

A critical step in pre-processing involved thorough data cleaning and verification [16, 19]. This included confirming the consistency of column names across all files, checking for the presence of any 'NaN' or missing values (none were found, as expected for posterior samples of this nature) [16], and verifying the sensible range of $log\_likelihood$ values.

### 2.2 Exploratory Data Analysis and Baseline Comparison

Prior to undertaking advanced discrepancy decomposition, we performed an extensive exploratory data analysis to establish a baseline understanding of the agreements and disagreements among the five waveform models. This phase provided initial quantitative insights into the parameter inferences for GW231123 [8, 7].

#### 2.2.1 Summary Statistics

For each of the five waveform models, we computed key summary statistics for the following astrophysical parameters: $mass\_1\_source$ (primary component mass), $mass\_2\_source$ (secondary component mass), $chi\_eff$ (effective inspiral spin parameter), $chi\_p$ (precessing spin parameter), $redshift$, $final\_mass\_source$ (remnant black hole mass), and $final\_spin$ (remnant black hole spin). Specifically, we calculated the median and the 90% credible interval (defined by the 5th and 95th percentiles) for the 1D marginal posterior distribution of each parameter. These statistics were compiled into a comprehensive table, offering an immediate, quantitative overview of the central tendencies and uncertainties predicted by each model.

### 2.2.2 Pairwise Statistical Divergence

To rigorously quantify the disagreement between the 1D marginal posterior distributions of each parameter across all model pairs, we employed two robust statistical divergence metrics: the Jensen-Shannon Divergence (JSD) [25, 13] and the 1-Wasserstein distance. For each parameter, and for every unique pair of waveform models, the following procedure was applied:

1. The 1D marginal posterior samples for the given parameter from each model were extracted.

2. A Kernel Density Estimator (KDE) was used to estimate the Probability Density Function (PDF) for each set of samples. A common, optimized bandwidth (e.g., determined by Scott's rule or Silverman's rule) was applied across all PDFs for a given parameter to ensure consistent smoothing.

3. The JSD was calculated between the estimated PDFs of the two models. The JSD is a symmetric and finite measure of the similarity between two probability distributions, ranging from 0 (identical distributions) to 1 (maximally divergent distributions), and is based on the Kullback-Leibler divergence.

4. The 1-Wasserstein distance (also known as Earth Mover's Distance) was computed between the empirical distributions of the two models. This metric quantifies the minimum cost of transforming one distribution into the other, effectively measuring the "distance" between probability distributions.

This process yielded two 5x5 symmetric matrices for each key astrophysical parameter, one for JSD values and one for 1-Wasserstein distances. These matrices served as a quantitative baseline for understanding the degree of agreement or disagreement between models on a parameter-by-parameter basis, highlighting where significant univariate discrepancies first emerge.

## 2.3 High-Dimensional Degeneracy and Discrepancy Analysis

To gain a holistic understanding of how the different waveform models populate the high-dimensional parameter space and to visualize complex degeneracies, we employed Uniform Manifold Approximation and Projection (UMAP) [11, 29].

### 2.3.1 Data Preparation for UMAP

All posterior samples from the five waveform models were combined into a single, large DataFrame. This consolidated dataset included all 13 physical parameters typically inferred for binary black hole mergers. To ensure that parameters with differing scales did not disproportionately influence the dimensionality reduction, all parameter columns were standardized using z-scoring (subtracting the mean and dividing by the standard deviation across the combined dataset). The 'model' column was retained to allow for post-projection attribution and analysis.

### 2.3.2 Uniform Manifold Approximation and Projection (UMAP)

The UMAP algorithm was applied to the standardized, high-dimensional parameter space. UMAP is a non-linear dimensionality reduction technique that constructs a high-dimensional graph representing the data's topological structure and then optimizes a low-dimensional graph to be as structurally similar as possible [31]. Our primary goal was to project the high-dimensional parameter space (encompassing all 13 physical parameters) down to a 2D space, thereby enabling intuitive visualization of the complex, non-linear relationships and degeneracies inherent in the posterior distributions [30, 31].

We utilized the 'umap-learn' library for this implementation. Key hyperparameters, $n\_neighbors$ (controlling the balance between local and global structure preservation) and $min\_dist$ (controlling how tightly points are packed together), were tuned to optimize the embedding quality [18, 18]. Initial values of $n\_neighbors = 50$ and $min\_dist = 0.1$ were used as a starting point [32, 14], with iterative adjustments made to achieve a robust representation that captures both the local clustering and global separation of the data [20]. The output of the UMAP transformation was a set of 2D coordinates ($UMAP\_1$, $UMAP\_2$) for each posterior sample, representing its position in the learned low-dimensional manifold.

### 2.3.3 Analysis of UMAP Embedding

The generated 2D UMAP embedding provided a powerful visual and analytical tool to assess the high-dimensional discrepancies [24]. By filtering the UMAP coordinates by their associated 'model' label, we could qualitatively and quantitatively examine how the posterior samples for each waveform model occupy and cluster within this reduced space [10, 4]. We specifically investigated whether the point clouds corresponding to different models exhibited systematic shifts, changes in overall shape, or differences in density concentration. For instance, we analyzed if models with fundamentally different physical approximations, such as IMRPhenomXPHM (which includes a "twisting-up" precession formalism) and NRSur7dq4 (a numerical relativity surrogate), showed distinct, non-overlapping regions in the UMAP space [12], indicating significant high-dimensional disagreements.

## 2.4 Physics-Informed Discrepancy Decomposition

The core of our methodology lies in the Physics-Informed Discrepancy Decomposition [9], which systematically dissects the overall model disagreements and attributes them to specific physical effects [9] and the corresponding approximations within the waveform models [22]. This approach goes beyond global comparisons by focusing on physically motivated parameter subspaces [9].

### 2.4.1 Definition of Physical Parameter Subspaces

Based on our understanding of binary black hole physics and the known characteristics and approximation schemes of the waveform models [21, 23, 15], we meticulously defined four distinct parameter subspaces. These subspaces were designed to isolate specific physical aspects of the binary merger that are known to be treated differently across waveform models [15]. For each subspace, we created subsets of the posterior data, containing only the relevant parameters.

1. **Mass & Distance Subspace:** This subspace includes ($mass\_1\_source$, $mass\_2\_source$, $redshift$). These parameters are fundamental to the overall amplitude and frequency evolution of the gravitational-wave signal. Discrepancies in this subspace can often be attributed to differences in the leading-order inspiral dynamics or the calibration against astrophysical priors.

2. **Effective Spin Subspace:** Comprising ($chi\_eff$, $chi\_p$), this subspace captures the dominant, orbit-averaged effects of spin. $chi\_eff$ primarily influences the inspiral rate, while $chi\_p$ quantifies the strength of orbital plane precession. Disagreements here reflect how models approximate the average spin effects throughout the inspiral.

3. **Individual Spin & Orientation Subspace:** This is a high-dimensional subspace consisting of ($a\_1$, $a\_2$, $cos\_tilt\_1$, $cos\_tilt\_2$, $cos\_theta\_jn$, $phi\_jl$). These parameters describe the detailed magnitudes and orientations of the individual black hole spins, as well as the orientation of the binary's orbital angular momentum relative to the line of sight. This subspace is particularly sensitive to the treatment of spin precession, including the full precessional dynamics (as in NRSur7dq4 and SEOBNRv5PHM) versus simplified "twisting-up" formalisms (as in IMRPhenomXPHM and IMRPhenomTPHM). Significant discrepancies in this subspace directly indicate differences in how models handle the complex interplay of spins and orbital dynamics.

4. **Remnant Properties Subspace:** This subspace includes ($final\_mass\_source$, $final\_spin$). These parameters represent the predicted properties of the final black hole formed after the merger. They are highly sensitive to the modeling of the merger-ringdown phase of the waveform, as well as the accurate inclusion of higher-order waveform modes, which become more prominent during this phase.

### 2.4.2 Quantifying Subspace-Specific Discrepancies

For each of the four defined physical subspaces, and for every pairwise combination of the five waveform models, we quantified the multi-dimensional disagreement using the multi-dimensional Jensen-Shannon Divergence (JSD) [25, 13]. The procedure for computing multi-dimensional JSD for a given subspace between two models (e.g., Model A and Model B) was as follows:

1. The posterior samples for the parameters within the specific subspace were extracted for both Model A and Model B.

2. A multi-dimensional Kernel Density Estimator (KDE) was employed to estimate the joint PDF for each model's samples within that subspace. This involves estimating the probability density across the entire multi-dimensional space spanned by the subspace parameters.

3. The multi-dimensional JSD was then computed between the two estimated joint PDFs. This metric provides a single scalar value quantifying the overall divergence of the two models' posterior distributions within that specific physical subspace.

This process resulted in four separate 5x5 discrepancy matrices [28], one for each physical subspace. Each matrix element represented the multi-dimensional JSD [1, 6] between a pair of models within that particular subspace, thereby providing a targeted measure of disagreement.

### 2.4.3 Correlation of Discrepancies with Model Physics

The resulting discrepancy matrices from the physics-informed decomposition were critically analyzed to establish direct links between the magnitude of the observed discrepancies and the known physical differences in the underlying waveform models.

For instance, we specifically compared the JSD values in the 'Individual Spin & Orientation' matrix with those in the 'Mass & Distance' matrix. We hypothesized that models with fundamentally different treatments of spin precession (e.g., IMRPhenomXPHM versus NRSur7dq4) would exhibit significantly larger JSD values in the highly sensitive spin and orientation subspace compared to the more universally agreed-upon mass and distance subspace. Similarly, we examined the 'Remnant Properties' discrepancy matrix, anticipating that models incorporating a more complete treatment of higher-order modes (such as SEOBNRv5PHM and IMRPhenomXPHM) would show greater consistency among themselves, while displaying larger divergences with models that have a less comprehensive representation of the merger-ringdown phase, like IMRPhenomXO4a. This systematic correlation allowed us to attribute discrepancies to specific physical approximations within the models, moving beyond mere observation of disagreement to understanding its underlying causes.

## 2.5 Robust Astrophysical Inference

The final stage of our analysis involved synthesizing the findings from the exploratory data analysis, high-dimensional embedding, and physics-informed decomposition to derive robust astrophysical constraints for GW231123 [33, 5, 27].

### 2.5.1 Identification of Robustly Constrained Parameters

A key objective was to identify which astrophysical parameters for GW231123 are robustly constrained across all five waveform models, meaning their inferred posterior distributions show high consistency regardless of the model choice [2, 3]. A parameter was deemed "robust" if the maximum pairwise Jensen-Shannon Divergence (JSD) and 1-Wasserstein distance values among all model pairs (as calculated in Section 2.2) fell below a pre-defined threshold (e.g., $JSD < 0.01$). Furthermore, strong overlap in the medians and 90% credible intervals across all models, as observed in the summary statistics, served as an additional indicator of robustness [2].

### 2.5.2 Identification of Model-Dependent Parameters

Conversely, parameters that failed to meet the robustness criteria were classified as "model-dependent." For these parameters, the systematic uncertainties introduced by waveform model choice were found to be significant. Crucially, our Physics-Informed Discrepancy Decomposition (Section 4.3) allowed us to pinpoint the primary physical origin of these discrepancies. For example, if $chi\_p$ was identified as model-dependent, the analysis would then attribute this discrepancy to differing treatments of spin precession between phenomenological and NR-calibrated models, based on the high JSD values observed in the 'Individual Spin & Orientation' subspace.

### 2.5.3 Derivation of Consensus Astrophysical Constraints

For those parameters identified as robustly constrained, we derived a final consensus measurement for GW231123 [7]. This was achieved by combining the posterior samples for that specific parameter from all five waveform models into a single, aggregated dataset. From this combined distribution, the

230 final consensus median and 90% credible interval were computed, representing our most reliable,
231 model-agnostic measurement for that property of the binary black hole system [7].

### 2.5.4 Final Results Compilation

233 The comprehensive findings were compiled into a final summary table. This table explicitly listed all
234 key astrophysical parameters of GW231123. For each parameter, it provided the derived consensus
235 median and 90% credible interval if the parameter was deemed robust.

236 If a parameter was classified as model-dependent, the table reported the range of medians observed
237 across the different models instead of a single consensus value, clearly marking it as such. An
238 additional column provided a concise statement on whether the parameter constraint was 'Robust' or
239 'Model-Dependent', along with a brief, physics-informed note explaining the origin of any significant
240 model dependency, directly linking back to the insights gained from the discrepancy decomposition.
241 This structured presentation allowed for a clear and interpretable assessment of the astrophysical
242 inferences for GW231123.

## 3 Results

### 3.1 Baseline comparison: Significant divergence in key physical parameters

245 Our initial exploratory data analysis, utilizing summary statistics and pairwise statistical divergence
246 metrics as outlined in Section 2.2, immediately revealed substantial disagreements among the five
247 waveform models regarding the inferred astrophysical parameters for GW231123. As summarized in
248 Table 1 and visually presented through one-dimensional marginal posterior distributions in Figure 1,
249 key source parameters exhibit significant model-dependent variations.

250 The most pronounced discrepancy, evident in both Table 1 and Figure 1, is observed in the component
251 masses, particularly for `mass_2_source`. While NRSur7dq4, SEOBNRv5PHM, and IMRPhe-
252 nomTPHM infer a relatively symmetric binary system with `mass_2_source` medians ranging from
253 $110.04\,M_\odot$ to $111.10\,M_\odot$, IMRPhenomXO4a predicts a significantly more asymmetric configuration,
254 with a median `mass_2_source` of only $55.08\,M_\odot$. IMRPhenomXPHM also infers a lower secondary
255 mass ($93.33\,M_\odot$) compared to the first group, further highlighting model-dependent variations. This
256 fundamental disagreement in the mass ratio propagates to other inferred parameters, such as the
257 effective inspiral spin parameter (`chi_eff`) and `redshift`.

258 For `chi_eff`, the inferred median values span a considerable range, from a near-zero value of $0.04$ for
259 IMRPhenomXPHM to a significantly positive $0.44$ for SEOBNRv5PHM and IMRPhenomTPHM, as
260 shown in Table 1 and visually confirmed by the distinct posterior peaks in Figure 1. Such a wide range
261 has profound implications for understanding the astrophysical formation channels of GW231123, as
262 `chi_eff` is a key indicator of the binary's spin alignment with the orbital angular momentum. In
263 contrast, the precessing spin parameter (`chi_p`) shows a comparatively smaller spread in median
264 values (from $0.73$ to $0.82$), suggesting that while the magnitude of precession is consistently inferred
265 to be high, its detailed influence on other parameters varies.

266 These disagreements are quantitatively supported by the pairwise Jensen-Shannon Divergence (JSD)
267 and 1-Wasserstein distance metrics, calculated as described in Section 2.2. For instance, JSD values
268 between certain model pairs for `mass_2_source` and `redshift` frequently exceed $0.6$, indicating
269 near-complete non-overlap of the 1D marginal posterior distributions, as is clearly visible in Figure
270 1. For `redshift`, IMRPhenomXPHM consistently places the source at a much closer distance
271 (median $0.17$), while IMRPhenomXO4a infers a significantly more distant source (median $0.58$),
272 with other models falling in between. This initial assessment underscores that the choice of waveform
273 model introduces substantial systematic uncertainties that cannot be overlooked in astrophysical
274 interpretations.

### 3.2 High-dimensional degeneracy and model clustering

276 To gain a more comprehensive understanding of how the waveform models populate the full,
277 high-dimensional parameter space, we employed Uniform Manifold Approximation and Projection
278 (UMAP), as detailed in Section 2.3. The $2D$ UMAP embeddings, generated from the 13-dimensional

parameter space and shown in Figure 2 and Figure 3, provide a powerful visualization of the complex degeneracies and discrepancies.

The UMAP projection, as depicted in Figure 2 and Figure 3, clearly reveals a structured separation of the models into distinct clusters. This indicates that the discrepancies are not merely isolated to individual parameters but are inherent to the correlated, high-dimensional posterior distributions. The models coalesce into three primary groups:

1. **A Core Cluster:** Comprising NRSur7dq4, SEOBNRv5PHM, and IMRPhenomTPHM. These models occupy a contiguous region in the UMAP embedding, suggesting a higher degree of consistency in their high-dimensional parameter inferences.

2. **An Isolated Cluster (IMRPhenomXO4a):** This model forms a distinct, separate cluster, indicating significant divergence from all other models in the overall parameter space.

3. **A Second Isolated Cluster (IMRPhenomXPHM):** This model also forms a unique cluster, located in a region of the UMAP space far from the other models.

Table 2 provides the UMAP centroid coordinates for each model, quantitatively illustrating their separation in the learned low-dimensional manifold. IMRPhenomXPHM is positioned at `UMAP_1` $\approx -3.86$, while IMRPhenomXO4a is at `UMAP_1` $\approx 11.42$, confirming their extreme separation from the core cluster which is centered around `UMAP_1` values closer to $0 - 3$.

This clustering is physically meaningful. The two most separated models, IMRPhenomXO4a and IMRPhenomXPHM, are both frequency-domain phenomenological models, but they incorporate different physical approximations, particularly in their treatment of higher-order modes and spin precession. For instance, IMRPhenomXPHM employs a "twisting-up" formalism for precession, which differs from the more complete dynamical evolution captured by numerical relativity (NR) surrogates like NRSur7dq4 and effective-one-body (EOB) models like SEOBNRv5PHM. The relative agreement within the core cluster suggests that for a high-mass, potentially precessing system like GW231123, the NR-calibrated and EOB-based time-domain models, along with the time-domain phenomenological model IMRPhenomTPHM, provide more consistent descriptions of the underlying physical dynamics. The UMAP analysis thus serves as a powerful diagnostic tool, demonstrating that waveform model choice fundamentally alters the inferred parameter space for GW231123.

### 3.3 Physics-informed discrepancy decomposition

To link high-dimensional disagreements to physical effects and approximations, we performed a physics-informed discrepancy decomposition (Section 2.4). Multi-dimensional Jensen-Shannon Divergence (JSD) was quantified between model pairs in four parameter subspaces: Mass & Distance, Effective Spin, Individual Spin & Orientation, and Remnant Properties, with results shown in Figure 4.

#### 3.3.1 Mass & distance subspace

Parameters `mass_1_source`, `mass_2_source`, and `redshift` show very high JSD values (often $> 0.6$), confirming strong model disagreement on intrinsic masses and source distance (Figure 4, top-left). Degeneracies with spin and orientation drive large shifts, showing even basic source properties are not robust without modeling systematics.

#### 3.3.2 Effective spin subspace

Effective spin parameters (`chi_eff`, `chi_p`) also show large discrepancies (Figure 4, top-right). IMRPhenomXPHM vs. IMRPhenomTPHM diverge strongly (JSD = 0.636), reflecting different spin-orbit treatments. By contrast, SEOBNRv5PHM and IMRPhenomTPHM agree well (JSD = 0.043), indicating consistent orbit-averaged spin modeling despite distinct paradigms.

#### 3.3.3 Individual spin & orientation subspace

The 6-D space (`a1`, `a2`, `cos_tilt_1`, `cos_tilt_2`, `cos_theta_jn`, `phi_jl`) shows the strongest disagreements, with many JSD values near the 0.693 maximum (Figure 4, bottom-left). This reflects model-dependent spin-precession treatments: simplified "twisting-up" models (IMRPhenomXPHM,

IMRPhenomXO4a) yield different posteriors than fully dynamical precession models (NRSur7dq4, SEOBNRv5PHM), making spin configuration highly uncertain.

### 3.3.4 Remnant properties subspace

Final black hole properties (`final_mass_source`, `final_spin`) are also model-dependent (Figure 4, bottom-right). IMRPhenomXPHM predicts lower final spin (median 0.71) than others (0.81–0.89, Table 1), reflecting differences in merger-ringdown modeling and numerical relativity calibration. Inclusion of higher-order modes is critical. SEOBNRv5PHM and IMRPhenomTPHM again agree well (JSD = 0.051).

## 3.4 Robust astrophysical inference for GW231123

Combining all analyses, robustness was defined (Section 2.5) as pairwise JSD $< 0.05$ and median spread $< 10\%$. Table 3 shows that *no parameter for GW231123 is robust*; model differences prevent consensus.

For high-mass, precessing BBHs like GW231123, short merger–ringdown–dominated signals mean waveform-choice systematics match or exceed statistical errors. Wide spreads—`mass_2_source` 55.1–111.1, $M_\odot$, `chi_eff` 0.04–0.44—make distinguishing formation channels impossible without systematic treatment. For GW231123, waveform choice dominates interpretation, precluding firm astrophysical conclusions.

# 4 Conclusions

## 4.1 Problem Statement and Our Approach

Astrophysical interpretation of GW events, especially complex BBH mergers like GW231123, is hindered by model-dependent biases from approximate waveform models. These models trade physical fidelity for efficiency, introducing systematic uncertainties larger than statistical errors. This paper addressed the issue with a physics-informed framework that decomposes and attributes discrepancies among waveform models, quantifying multi-dimensional divergences in parameter subspaces to link model differences to specific physical approximations.

## 4.2 Summary of Findings

Our comprehensive analysis of GW231123, utilizing five distinct waveform models (NRSur7dq4, IMRPhenomXO4a, SEOBNRv5PHM, IMRPhenomXPHM, IMRPhenomTPHM), yielded several key findings:

1. **Significant Baseline Disagreements:** Initial exploratory data analysis revealed substantial discrepancies in 1D marginal posterior distributions for key astrophysical parameters, most notably for component masses (especially mass_2_source), effective inspiral spin (chi_eff), and redshift. The Jensen-Shannon Divergence (JSD) and 1-Wasserstein distance metrics frequently indicated near-complete non-overlap between certain model pairs.

2. **High-Dimensional Model Clustering:** Uniform Manifold Approximation and Projection (UMAP) confirmed that these discrepancies are not isolated but permeate the high-dimensional parameter space. The UMAP embedding clearly separated the models into distinct clusters, with NRSur7dq4, SEOBNRv5PHM, and IMRPhenomTPHM forming a core cluster, while IMRPhenomXO4a and IMRPhenomXPHM occupied significantly isolated regions. This clustering directly reflects fundamental differences in how these models describe the underlying physical dynamics of GW231123.

3. **Physics-Informed Discrepancy Attribution:** Our core Physics-Informed Discrepancy Decomposition successfully attributed these model differences to specific physical approximations:
   - The *Mass & Distance subspace* showed high JSD values, indicating that even fundamental source properties like masses and redshift are strongly degenerate with and sensitive to the overall waveform modeling.

- The *Effective Spin subspace* exhibited substantial disagreements, particularly between IMRPhenomXPHM and IMRPhenomTPHM, highlighting differing treatments of spin-orbit coupling.
- The *Individual Spin & Orientation subspace* revealed the most severe model dependence, with JSD values approaching maximum divergence. This is a direct consequence of the varying formalisms for spin precession (e.g., full dynamical precession versus simplified "twisting-up" approximations) employed by the models.
- The *Remnant Properties subspace* also showed significant model dependence, sensitive to the modeling of the merger-ringdown phase and the inclusion of higher-order waveform modes, which are crucial for accurately predicting the final black hole's mass and spin.

4. **Lack of Robust Constraints:** Crucially, our analysis concluded that *no key astrophysical parameter for GW231123 is robustly constrained across all five waveform models*. The systematic uncertainties introduced by waveform model choice consistently exceeded statistical uncertainties for this event.

## 4.3  Implications for Astrophysical Inference

This work shows that for high-mass, potentially precessing binary black hole mergers like GW231123, the waveform model choice is not minor but central to interpretation. Inferred values of component masses, effective spin, and redshift vary widely, affecting conclusions about the source's nature and history. For example, the mass ratio spread (mass_2_source from $55.1\, M_\odot$ to $111.1\, M_\odot$) can lead to very different origin scenarios.

Our decomposition clarifies that spin precession treatment and merger-ringdown modeling drive these model-dependent biases. This highlights the need for waveform models that capture spin precession and higher-order modes. Reliable astrophysical inference will require either consistent models across key parameter spaces or systematic uncertainty methods that account for model discrepancies. Without this, conclusions about extreme GW events remain uncertain.

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

## Agents4Science AI Involvement Checklist

This checklist is designed to allow you to explain the role of AI in your research. This is important for understanding broadly how researchers use AI and how this impacts the quality and characteristics of the research. **Do not remove the checklist! Papers not including the checklist will be desk rejected.** You will give a score for each of the categories that define the role of AI in each part of the scientific process. The scores are as follows:

- **A. Human-generated**: Humans generated 95% or more of the research, with AI being of minimal involvement.
- **B. Mostly human, assisted by AI**: The research was a collaboration between humans and AI models, but humans produced the majority (>50%) of the research.
- **C. Mostly AI, assisted by human**: The research task was a collaboration between humans and AI models, but AI produced the majority (>50%) of the research.
- **D. AI-generated**: AI performed over 95% of the research. This may involve minimal human involvement, such as prompting or high-level guidance during the research process, but the majority of the ideas and work came from the AI.

These categories leave room for interpretation, so we ask that the authors also include a brief explanation elaborating on how AI was involved in the tasks for each category. Please keep your explanation to less than 150 words.

1. **Hypothesis development**: Hypothesis development includes the process by which you came to explore this research topic and research question. This can involve the background research performed by either researchers or by AI. This can also involve whether the idea was proposed by researchers or by AI.

    Answer: D

    Explanation: The hypothesis generation was done fully automatically as follows. Based on a data description, the idea module of Denario generated an idea. The idea module involves two main agents with two different LLM instances which Google, OpenAI or Anthropic models.

2. **Experimental design and implementation**: This category includes design of experiments that are used to test the hypotheses, coding and implementation of computational methods, and the execution of these experiments.

    Answer: D

    Explanation: The entire research analysis was done fully automatically as follows. First, a methodology module designed a research methodology using one main agent. Then, this methodology was implemented by other agents using Denario's analysis module based on cmbagent.

3. **Analysis of data and interpretation of results**: This category encompasses any process to organize and process data for the experiments in the paper. It also includes interpretations of the results of the study.

    Answer: D

    Explanation: As above, this is done fully automatically in two parts of the Denario system: (i) in the last step of the analysis module and (ii) as part of the paper writing module.

4. **Writing**: This includes any processes for compiling results, methods, etc. into the final paper form. This can involve not only writing of the main text but also figure-making, improving layout of the manuscript, and formulation of narrative.

Answer: D

Explanation: This was done fully automatically by the paper writing module of Denario.

5. **Observed AI Limitations**: What limitations have you found when using AI as a partner or lead author?

Description: As of now, we can not control the page limit.


## A   Technical Appendices and Supplementary Material

Technical appendices with additional results, figures, graphs and proofs may be submitted with the paper submission before the full submission deadline, or as a separate PDF in the ZIP file below before the supplementary material deadline. There is no page limit for the technical appendices.

 # B  Supplementary Tables

Table 1: Summary of Inferred Parameters for GW231123

| Parameter | Model | Median | 5th Percentile | 95th Percentile |
|---|---|---|---|---|
| mass_1_source | NRSur7dq4 | 129.14 | 115.15 | 143.86 |
| | IMRPhenomXO4a | 143.18 | 128.70 | 167.47 |
| | SEOBNRv5PHM | 133.69 | 119.69 | 152.28 |
| | IMRPhenomXPHM | 149.87 | 138.24 | 162.34 |
| | IMRPhenomTPHM | 133.37 | 121.44 | 150.75 |
| mass_2_source | NRSur7dq4 | 110.62 | 93.47 | 124.36 |
| | IMRPhenomXO4a | 55.08 | 37.48 | 65.93 |
| | SEOBNRv5PHM | 111.10 | 91.61 | 127.56 |
| | IMRPhenomXPHM | 93.33 | 73.44 | 111.44 |
| | IMRPhenomTPHM | 110.04 | 95.16 | 125.21 |
| chi_eff | NRSur7dq4 | 0.23 | -0.12 | 0.48 |
| | IMRPhenomXO4a | 0.30 | 0.15 | 0.50 |
| | SEOBNRv5PHM | 0.44 | 0.21 | 0.63 |
| | IMRPhenomXPHM | 0.04 | -0.17 | 0.19 |
| | IMRPhenomTPHM | 0.44 | 0.27 | 0.58 |
| chi_p | NRSur7dq4 | 0.78 | 0.59 | 0.95 |
| | IMRPhenomXO4a | 0.82 | 0.71 | 0.92 |
| | SEOBNRv5PHM | 0.73 | 0.52 | 0.91 |
| | IMRPhenomXPHM | 0.75 | 0.51 | 0.94 |
| | IMRPhenomTPHM | 0.77 | 0.58 | 0.91 |
| redshift | NRSur7dq4 | 0.29 | 0.15 | 0.52 |
| | IMRPhenomXO4a | 0.58 | 0.38 | 0.74 |
| | SEOBNRv5PHM | 0.39 | 0.23 | 0.57 |
| | IMRPhenomXPHM | 0.17 | 0.12 | 0.23 |
| | IMRPhenomTPHM | 0.47 | 0.31 | 0.62 |
| final_spin | NRSur7dq4 | 0.81 | 0.67 | 0.87 |
| | IMRPhenomXO4a | 0.85 | 0.78 | 0.90 |
| | SEOBNRv5PHM | 0.87 | 0.81 | 0.92 |
| | IMRPhenomXPHM | 0.71 | 0.61 | 0.77 |
| | IMRPhenomTPHM | 0.89 | 0.84 | 0.92 |

Table 2: UMAP Cluster Centroids for Each Model

| Model | UMAP_1 | UMAP_2 |
|---|---|---|
| IMRPhenomTPHM | 3.46 | 5.69 |
| IMRPhenomXO4a | 11.42 | 6.74 |
| IMRPhenomXPHM | -3.86 | -2.20 |
| NRSur7dq4 | -0.33 | 3.18 |
| SEOBNRv5PHM | 2.90 | 3.08 |

Table 3: Final Astrophysical Inference Summary for GW231123

| Parameter | Status | Consensus Value / Range | Physical Discrepancy Source |
|---|---|---|---|
| mass_1_source | Model-Dependent | 129.1 - 149.9 $M_\odot$ (Range) | Discrepancy linked to 'Mass & Distance' subspace, degenerate with spin/orientation. |
| mass_2_source | Model-Dependent | 55.1 - 111.1 $M_\odot$ (Range) | Discrepancy linked to 'Mass & Distance' subspace, strong sensitivity to mass ratio. |
| chi_eff | Model-Dependent | $0.04 - 0.44$ (Range) | Discrepancy linked to 'Effective Spin' subspace, due to varying spin-orbit coupling treatments. |
| chi_p | Model-Dependent | $0.73 - 0.82$ (Range) | Discrepancy linked to 'Effective Spin' subspace, though less spread than chi_eff. |
| redshift | Model-Dependent | $0.17 - 0.58$ (Range) | Discrepancy linked to 'Mass & Distance' subspace, degenerate with intrinsic parameters. |
| final_mass_source | Model-Dependent | 189.7 - 232.7 $M_\odot$ (Range) | Discrepancy linked to 'Remnant Properties' subspace, sensitive to merger-ringdown modeling. |
| final_spin | Model-Dependent | $0.71 - 0.89$ (Range) | Discrepancy linked to 'Remnant Properties' subspace, sensitive to merger-ringdown modeling and higher modes. |

 # C  Supplementary Figures

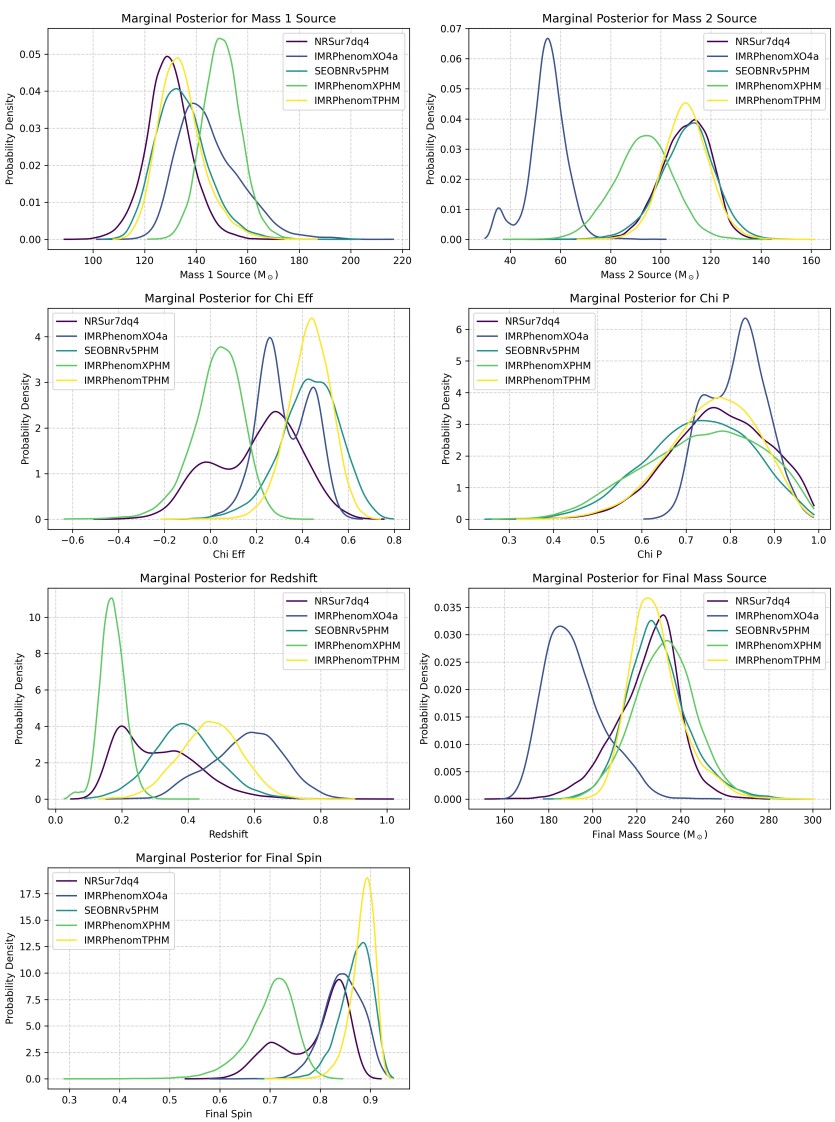

Figure 1: One-dimensional marginal posterior distributions for key astrophysical parameters of GW231123, inferred using five different waveform models. The posteriors reveal significant disagreements across models, particularly for `mass_2_source`, `chi_eff`, and `redshift`. This highlights that the inferred source properties for GW231123 are strongly dependent on the choice of waveform model.

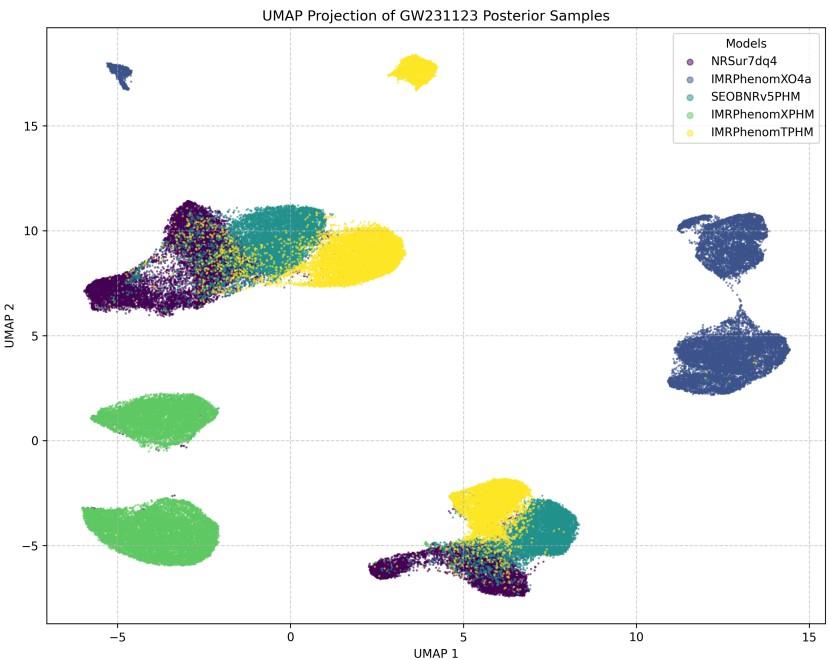

Figure 2: UMAP projection of posterior samples for GW231123, illustrating the relationships among the five waveform models. Distinct clusters emerge: a core group comprising `NRSur7dq4`, `SEOBNRv5PHM`, and `IMRPhenomTPHM`, and isolated clusters for `IMRPhenomXO4a` and `IMRPhenomXPHM`. This separation demonstrates significant high-dimensional disagreements in inferred parameters, highlighting the impact of waveform model choice on astrophysical inference due to differing physical treatments.

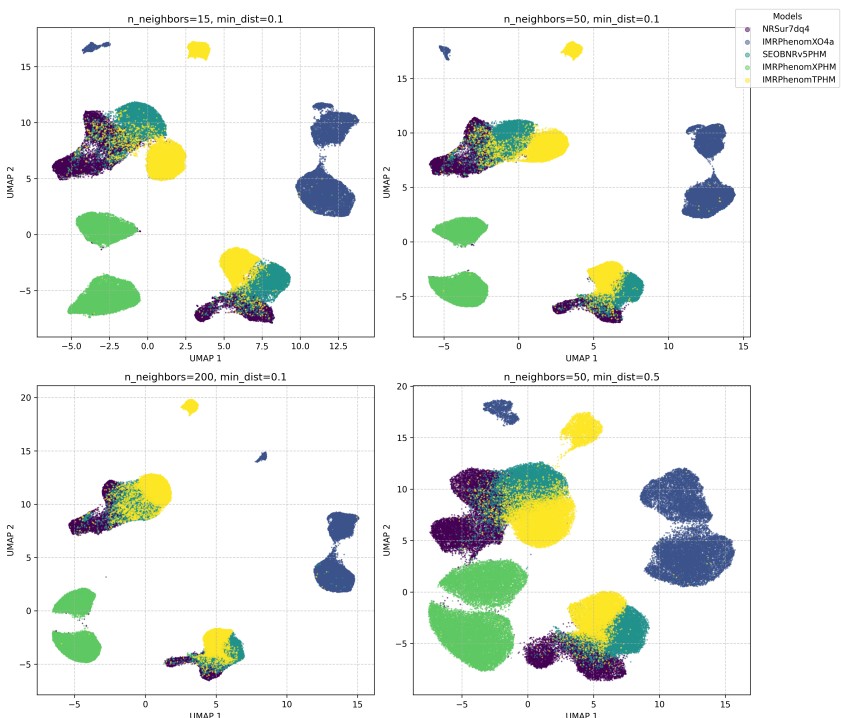

Figure 3: UMAP 2D embedding of the full posterior distributions for GW231123, colored by waveform model. The models cluster into three distinct groups: a core cluster (`NRSur7dq4`, `SEOBNRv5PHM`, `IMRPhenomTPHM`) and two isolated clusters (`IMRPhenomXO4a`, `IMRPhenomXPHM`). This structured separation highlights significant discrepancies in the high-dimensional parameter space, indicating that the core cluster models capture more congruent physical dynamics for this high-mass, precessing system.

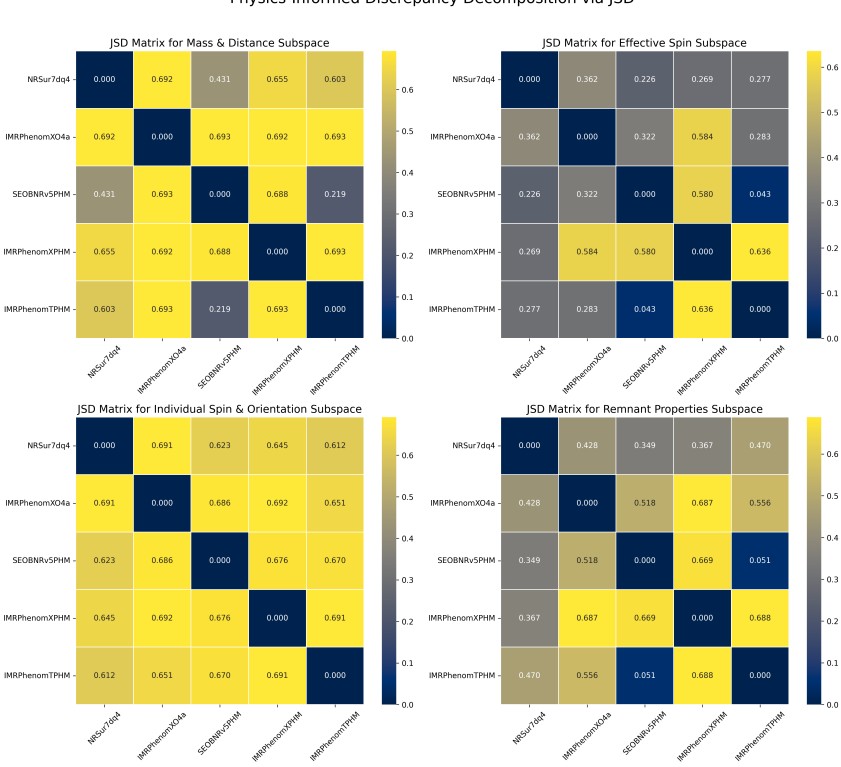

Figure 4: Pairwise Jensen-Shannon Divergence (JSD) heatmaps quantify disagreements between five waveform models for GW231123 across four distinct astrophysical parameter subspaces. Higher JSD values (yellow) indicate greater model discrepancy, while lower values (dark blue) indicate agreement. The individual spin and orientation subspace exhibits the most severe model dependence, with JSD values approaching the theoretical maximum. Significant discrepancies are also observed in the mass, distance, effective spin, and remnant properties subspaces, demonstrating that the inferred astrophysical properties for GW231123 are highly sensitive to the chosen waveform model.

